# Spectrum of antibiotic resistant bacteria and fungi isolated from chronically infected wounds in a rural district hospital in Ghana

**Ralf Krumkamp**[1,2☯], **Kwabena Oppong**[3☯], **Benedikt Hogan**[1], **Ricardo Strauss**[1],
**Hagen Frickmann**[4], **Charity Wiafe-Akenten**[🆔][3], **Kennedy G. Boahen**[3], **Volker Rickerts**[5],
**Ilka McCormick Smith**[5], **Uwe Groß**[6], **Marco Schulze**[6], **Anna Jaeger**[1], **Ulrike Loderstädt**[1],
**Nimako Sarpong**[3], **Ellis Owusu-Dabo**[3], **Jürgen May**[1], **Denise Dekker**[🆔][1,2]*

**1** Department of Infectious Disease Epidemiology, Bernhard Nocht Institute for Tropical Medicine (BNITM), Hamburg, Germany, **2** German Centre for Infection Research (DZIF), Hamburg-Lübeck-Borstel-Riems, Germany, **3** Kumasi Centre for Collaborative Research in Tropical Medicine (KCCR), Kumasi, Ghana, **4** Department for Tropical Medicine at the Bernhard Nocht Institute, Bundeswehr Hospital Hamburg, Germany, **5** Department Mycotic and Parasitic Agents and Mycobacteria, Robert Koch Institute (RKI), Berlin, Germany, **6** Institute for Medical Microbiology, University Medical Center Göttingen (UMG), Göttingen, Germany

☯ These authors contributed equally to this work.
* dekker@bnitm.de

**Data Availability Statement:** All relevant data are within the manuscript

## Abstract

### Background

Chronic infected wounds are generally difficult to manage and treatment can be particularly challenging in resource-limited settings where diagnostic testing is not readily available. In this study, the epidemiology of microbial pathogens in chronically infected wounds in rural Ghana was assessed to support therapeutic choices for physicians.

### Methods

Culture-based bacterial diagnostics including antimicrobial resistance testing were performed on samples collected from patients with chronic wounds at a hospital in Asante Akim North Municipality, Ghana. Fungal detection was performed by broad-range fungal PCR and sequencing of amplicons.

### Results

In total, 105 patients were enrolled in the study, from which 207 potential bacterial pathogens were isolated. *Enterobacteriaceae* (n = 84, 41%) constituted the most frequently isolated group of pathogens. On species level, *Pseudomonas aeruginosa* (n = 50, 24%) and *Staphylococcus aureus* (n = 28, 14%) were predominant. High resistance rates were documented, comprising 29% methicillin resistance in *S. aureus* as well as resistance to 3[rd] generation cephalosporins and fluoroquinolones in 33% and 58% of *Enterobacteriaceae*, respectively. One *P. aeruginosa* strain with carbapenem resistance was identified. The most frequently detected fungi were *Candida tropicalis*.

**Funding:** The author(s) received no specific funding for this work.

**Competing interests:** The authors have declared that no competing interests exist.

## Conclusions

The pathogen distribution in chronic wounds in rural Ghana matched the internationally observed patterns with a predominance of *P. aeruginosa* and *S. aureus*. Very high resistance rates discourage antibiotic therapy but suggest an urgent need for microbiological diagnostic approaches, including antimicrobial resistance testing to guide the management of patients with chronic wounds in Ghana.

## Introduction

Chronic wound infections related to injuries from daily life activities, e.g. farming, are common medical problems in sub-Saharan Africa (SSA) [1]. Chronicity is often a result of an underlying medical condition such as diabetes, blood flow disorders or due to late presentation to the clinic. Long persisting infected wounds cause morbidity and suffering and are typically associated with large expenditures, e.g. on medication and affect economic productivity [2].

In resource-limited countries with inadequate diagnostic facilities, the spectrum of micro-organisms causing such infections and their antibiotic resistance patterns are only superficially understood [3–6]. Pathogen-specific treatment for severe wounds is difficult if the disease-causing agent(s) remain(s) unknown. In addition, treatment success is under serious threat due to the rise of antimicrobial resistances [7]. Poor quality medicine and (over)- use of antibiotics without being able to consider the antimicrobial sensitivity patterns of causative pathogens does not only lead to treatment failures but also foster the development of antibiotic resistance further. These limitations hamper specific treatment, which especially applies to chronic wounds, for which standard treatment strategies are poorly defined and ineffective.

Besides bacteria, fungi are widespread in the environment or even part of the normal flora including *Aspergillus fumigatus* and *Candida albicans* among others. Such fungi may be part of polymicrobial communities on wounds. They may be associated with delayed healing especially in patient with underlying conditions including Diabetes mellitus. In addition, specific fungal pathogens may cause chronic skin infections, including mycetoma and chromoblasto-mycosis often manifesting as slow-healing wounds [8,9].

This study aims to investigate the microbial composition (both bacterial and fungal) of infected chronic wounds in adults presenting to the Agogo Presbyterian Hospital (APH) in rural Ghana. The study will provide information on bacterial strains and antibiotic resistance associated with chronic wounds. This information will help to improve patient care through providing evidence-based recommendations for treatment and management of chronic wounds.

## Materials and methods

### Study site and study participants

The study was conducted at the Outpatient Department of the APH, in the Asante Akim North Municipality of Ghana. Asante Akim North district is one of 21 in the Ashanti region. Injuries related to farm work and other outdoor activities and resulting chronic wound infections (due to delayed presentation) are frequent neglected medical problems in rural areas of Ghana.

Patients aged ≥15 years with an infected wound (at least one of the signs and symptoms of infection: purulent discharge from wound/pain or tenderness, localised swelling or redness/

heat, loss of function (functio laesa)), which has failed to proceed through a reparative process over a period of one month, with or without antibiotic treatment, were eligible for enrolment. The diameter of the wound was measured using a single-use paper tape measure. Wounds were graded from one to six (grade 1 = ulcer of epidermis or dermis, grade 2 = ulcer involving subcutis, grade 3 = involvement of fascia, grade 4 = involvement of muscles, grade 5 = bones affected, grade 6 = visceral cavity affected). Excluded were patients with burn-, surgical- bite- or Buruli ulcer associated wounds in order to minimise associations of particular types of wounds with specific pathogens.

## Ethical considerations

The Committee on Human Research, Publications and Ethics, School of Medical Science, Kwame Nkrumah University of Science and Technology in Kumasi, Ghana, approved this study (approval number CHRPE/AP/078/16). Study participants were informed about the purpose of this study and the study procedures. Written informed consent was obtained before enrolment.

## Sample collection

For sample collection two sterile swabs (ESwab$^{TM}$, COPAN Diagnostics, Murrieta, USA), one with and one without transport medium, were used. Necrotic material was removed with a sterile cotton swab (CLASSIQSwab$^{TM}$, COPAN Diagnostics). For sample collection, the wound was cleaned with sterile 0.9% saline. Following this, the active part of the wound below the necrotic tissue at the edge of the wound and the wound base was swabbed. For this purpose, the swab was rolled deep into the wound. When there was more than one wound at the same location, the largest wound was sampled.

Within six hours, the samples were transported in a cool box to the microbiology laboratory of the Kumasi Centre for Collaborative Research in Tropical Medicine, Ghana.

## Bacterial detection and identification

The swab with transport medium was streaked on Columbia agar enriched with 5% sheep blood, Columbia CNA agar with 5% sheep blood and MacConkey III (Oxoid, Hampshire, United Kingdom). Aerobic blood agar was incubated in $CO_2$. MacConkey and CNA agar were incubated in normal atmosphere at 35–37˚C. All plates were incubated for 18–24 hours. CNA agar was incubated until positive or for a maximum period of five days. Bacterial strains were identified by colony morphology, Gram stain and standard biochemical methods and stored in microbanks at -80˚C until transportation on dry ice to the Bernhard Nocht Institute for Tropical Medicine in Hamburg, Germany for further analysis.

Environmental bacteria and bacteria belonging to the skin microbiota (e.g. coagulase-negative staphylococci, *Micrococcus* spp., coryneform bacteria or *Bacillus* spp. other than *Bacillus anthracis*) were classified as contaminants. At the Institute for Medical Microbiology in Göttingen, Germany, species identification was confirmed using the MALDI Biotyper 3.0 (Bruker Daltonics, Bremen, Germany).

MALDI-TOF-MS measurements were carried out according to the MALDI Biotyper standard method (Bruker Daltonics, Bremen, Germany) using smear preparations. Species identification was confirmed in duplicate preparations from Columbia blood agar (bioMérieux, Marcy-l'Étoile, France). During the analysis, 600 spectra in a mass range between 2 and 20 kDa were collected in 100-shot steps on an Autoflex III system and summarized. Identification score values ≥2.00 achieved with MALDI Biotyper (database version 2016) were considered correct.

## Antibiotic susceptibility testing

Susceptibility to locally-available antibiotics was tested by the disk diffusion method and interpreted following the European Committee on Antimicrobial Susceptibility Testing (EUCAST) guidelines v.6.0 (http://www.eucast.org). Antibiotics of choice for bacterial isolates are illustrated in Table 2. Quality control of susceptibility testing was performed according to EUCAST (QC table v.5). At the University Medical Centre, Göttingen, Germany antibiotic susceptibility testing was confirmed using the automated VITEK 2 system (bio-Mérieux, Marcy-l'Étoile, France). Quality control was performed with the following reference strains: *Pseudomonas aeruginosa* ATCC 27853, *Staphylococcus aureus* ATCC 29213 and *Escherichia coli* 2 ATCC 5922.

## Fungal detection and identification

DNA was extracted from the swab (without transport medium) within 12 hours according to manufacturer's guidelines using the BIOstic Bacteremia DNA Isolation kit (QIAGEN, Hilden, Germany). DNA extracts were stored at -80˚C until polymerase chain reaction (PCR) amplification for clinically important fungi. Fungal DNA was amplified using a broad-range fungal PCR targeting the 28S rRNA gene (primer 28S10f: GACATGGGTTAGTCGATCCTA; 28S12r: CCTTATCTACATTRTTCTATCAAC) using Eva Green with melt curve analysis in an ABI 7500 qPCR machine as described previously [10]. Samples were tested in duplicate. Positive fungal PCR was defined as amplification of DNA of identical melt curves in both duplicates with a threshold cycle below 45. Template controls (n = 8) were included in each run to document potential contamination with fungal DNA. Amplicons of PCR positive samples were sequenced by Sanger sequencing. Fungi were identified by BLAST search using Genbank. Identification of fungi required identity >98% over the amplicon length.

In addition, a qPCR assay targeting the human 18S rRNA was performed to document successful sampling and DNA extraction. Furthermore, potential inhibition of the PCR reaction was screened by an internal amplification control, amplifying an artificial plasmid.

## Statistical analyses

Categorical variables were described using frequencies and their proportion and continuous variables using the median and the interquartile range (IQR). Prevalence ratios (PRs) along with their 95%-confidence intervals (CIs) were calculated to show associations between dichotomous variables. Missing values were excluded from the analysis; hence, in some calculations the denominator differs. All analyses were conducted using Stata Statistical Software 14 (StataCorp LLC, College Station, TX).

## Results

### Study group

One hundred and five outpatients were enrolled between January 2016 and November 2016. Characteristics of study patients are summarised in Table 1. Study participants had a median age of 54 years (IQR: 36–69) and half of the patients (n = 49; 47%) were females. From 47 (45%) patients, underlying diseases were reported, with the most frequent being hypertension (n = 27; 26%) and diabetes mellitus (n = 22; 21%). The majority of wounds were located on the leg (n = 76; 72%) followed by the foot (n = 13; 12%). Most wounds were ≥7 months old (n = 48; 46%), followed by below 8 weeks (n = 33; 31%). Wounds had a median area size of 15 cm$^2$ (IQR: 5–44. Patient's wounds showed the following characteristics: pain (n = 97, 92%), indurated border (n = 56; 53%), presence of pus (n = 43; 41%), functio laesa (n = 34; 32%) and local swelling (n = 15; 14%). In 55 (53%) patients the grade of the wound was stage two or

**Table 1. Patient and wound characteristics for Gram-positive and Gram-negative bacteria.**

| Characteristic | All (N = 105) | Gram-negative bacteria present (N = 85) | Only Gram-positive bacteria (N = 20) |
|---|---|---|---|
| Female sex [n (%)] | 49 (47) | 6 (30) | 43 (51) |
| Age (years) of patient [median (IQR)] | 54 (36–69) | 49 (24–59) | 56 (38–69) |
| Previous antibiotic use [n (%)] | 70 (67) | 57 (67) | 13 (65) |
| Area size (cm$^2$) of wound [median (IQR)] | 15 (5–44) | 5 (5–15) | 15 (5–44) |
| Duration of wound [n (%)] | | | |
| <8 weeks | 33 (31) | 28 (33) | 5 (25) |
| 9 weeks–6 months | 24 (23) | 20 (24) | 4 (20) |
| ≥7 months | 48 (46) | 37 (44) | 11 (55) |
| Location of wound [n (%)] | | | |
| leg/ankle | 76 (72) | 63 (74) | 13 (65) |
| foot | 13 (12) | 10 (12) | 3 (15) |
| other | 16 (15) | 12 (14) | 4 (20) |
| Aetiology of wound [n (%)] | | | |
| cut | 36 (34) | 29 (34) | 7 (35) |
| abscess/tissue swelling | 30 (29) | 25 (29) | 5 (25) |
| trauma/accidents | 12 (11) | 10 (12) | 2 (10) |
| other | 27 (26) | 21 (25) | 6 (30) |

higher. Most common reported causes of the wound were cuts (e.g. by a piece of wood or sharp object from the environment) (n = 36; 34%) followed by abscesses/swollen tissues (n = 30; 29%), traumata/accidents (n = 12; 11%) and blisters (n = 11; 10%). In 70 (67%) of the patients the wound was previously treated with antibiotics and in 38 (36%) with traditional medicine. Patients using herbal treatment were less likely to take antibiotics (PR = 0.7; 95%-CI: 0.5–1.0). Patients who used herbal treatment were more likely to have wounds for 9 weeks to 6 months (PR = 1.8; 95%-CI = 1.2–2.6).

## Bacterial isolates from chronic wounds

In total, 207 potential bacterial pathogens were isolated from 105 wounds and polymicrobial infections were frequent. In 7 (7%) of the patients no bacterial pathogen was isolated. In 34 (32%) one, in 31 (30%) two, in 21 (20%) three and in 12 (11%) patients four, bacterial pathogens were isolated. Wounds ≥5 cm$^2$ were more likely to carry multiple pathogens (PR = 1.5; 95%-CI: 1.0–2.3). The detected potential pathogens are listed in S1 Table. The majority of isolates were Gram-negative (n = 143; 69%) bacteria. Most were *Enterobacteriaceae* (n = 84; 41%) and *P. aeruginosa* (n = 50; 24%) followed by *S. aureus* (n = 28; 14%). In general, the bacterial composition was comparable between mono- and polymicrobial infections (Fig 1). The most frequently detected pathogen in mono- and polymicrobial infections was *P. aeruginosa*. In patients with one or two isolates detected, *S. aureus* was the second most common pathogen found. Patient and wound characteristics stratified by the presence or absence of Gram-negative bacteria are summarised in Table 1, indicating that characteristics were comparable between both groups. However, the size of wounds that contained Gram-positive bacteria were larger (median = 15 cm$^2$; IQR: 5–44) compared to wounds from which only Gram-negative bacteria were isolated (median = 5 cm$^2$; IQR: 5–15).

## Antibiotic susceptibility testing

Antibiotic susceptibility results of the most common bacterial species using the VITEK 2 system are shown in Table 2 and Fig 2. Among all *S. aureus*, 29% (n/N = 8/28) were methicillin

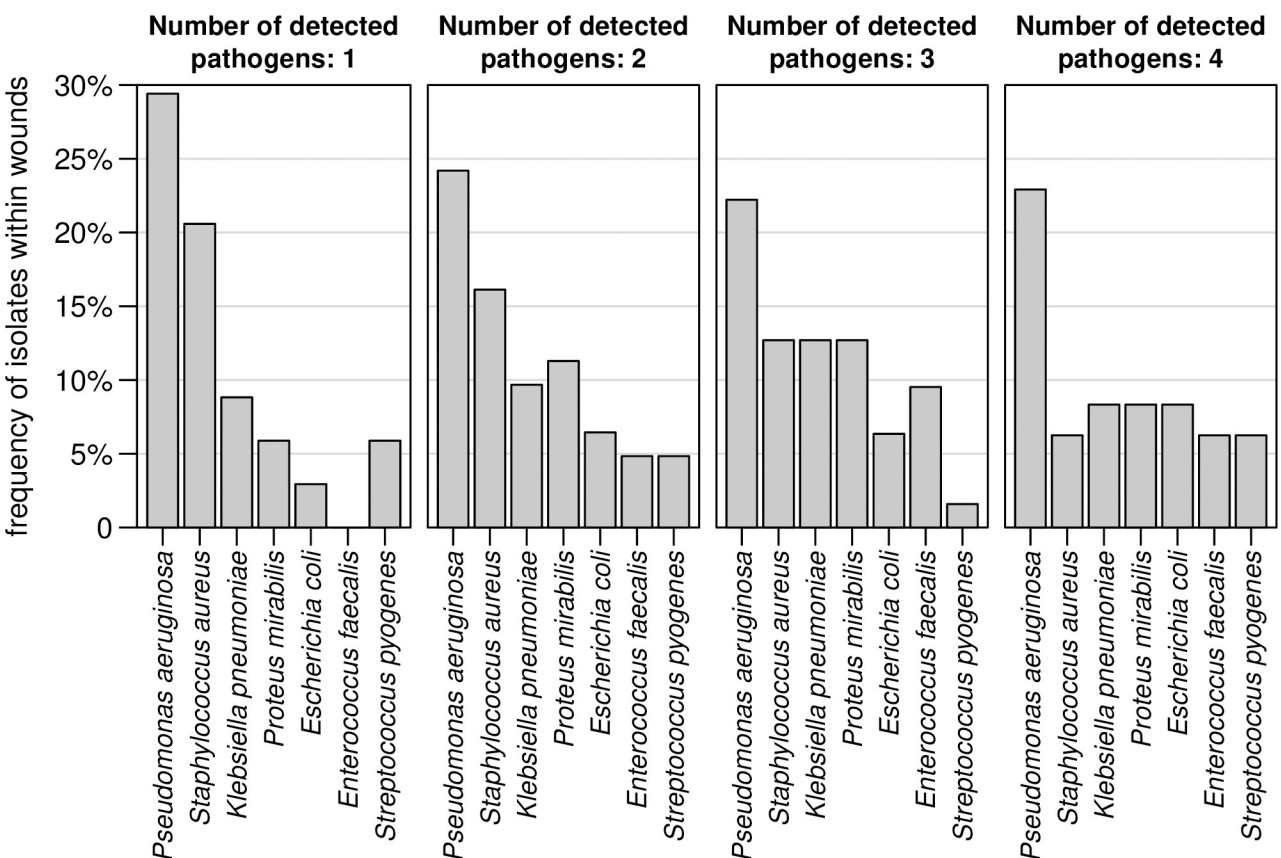

**Fig 1. Pathogen distribution in mono- and polymicrobial bacterial wound infection.** The different plots represent frequencies from wounds where 1, 2, 3 or 4 pathogens were detected.

**Table 2. Antibiotic resistance [n/N (%)] for selected bacterial species in chronic wound infections, Ghana, 2016.**

| Isolate | Enterococcus faecalis | Escherichia coli | Klebsiella pneumoniae | Proteus mirabilis | Staphylococcus aureus | Streptococcus pyogenes |
|---------|-----------------------|------------------|------------------------|-------------------|------------------------|-------------------------|
| **GEN** | - | 2/12 (17) | 6/21 (29) | 1/20 (5) | 1/28 (4) | - |
| **CIP** | - | 7/12 (58) | 3/21 (14) | 1/20 (5) | 0/28 (0) | - |
| **CAZ** | - | 3/12 (25) | 5/21 (24) | 0/21 (0) | - | - |
| **MER** | - | 0/12 (0) | 0/21 (0) | 0/20 (0) | - | - |
| **PEN** | 12/12 (100) | - | - | - | 28/28 (100) | 0/5 (0) |
| **ERY** | 12/12 (100) | - | - | - | 2/28 (7) | 2/5 (40) |
| **CLI** | 12/12 (100) | - | - | - | 1/28 (4) | 0/5 (0) |
| **SXT** | 12/12 (58) | 11/12 (92) | 10/21 (48) | 9/21 (43) | 11/28 (39) | - |
| **TET** | 12/12 (100) | - | - | 1/1 (100) | 16/28 (57) | - |
| **CHL** | 4/9 (44) | 10/12 (83) | 12/13 (92) | 9/14 (64) | 15/21 (71) | 5/8 (63) |
| **AMP** | 0/12 (0) | 12/12 (100) | 21/21 (100) | 19/21 (90) | - | 5/5 (100) |
| **SAM** | 0/12 (0) | 12/12 (100) | 21/21 (100) | 19/21 (90) | 8/28 (29) | - |
| **CXM** | - | 12/12 (100) | 21/21 (100) | 21/21 (100) | 8/28 (29) | - |
| **CRO** | - | 3/11 (27) | 4/21 (19) | 0/19 (0) | - | - |

GEN, gentamicin; CIP, ciprofloxacin; CAZ, ceftazidime; MER, meropenem; PEN, penicillin; ERY, erythromycin; CLI, clindamycin; SXT, trimethoprim/sulfamethoxazole; TET, tetracycline; CHL, chloramphenicol; AMP, ampicillin; SAM, ampicillin/sulbactam; CXM, cefuroxime; CRO, ceftriaxone.

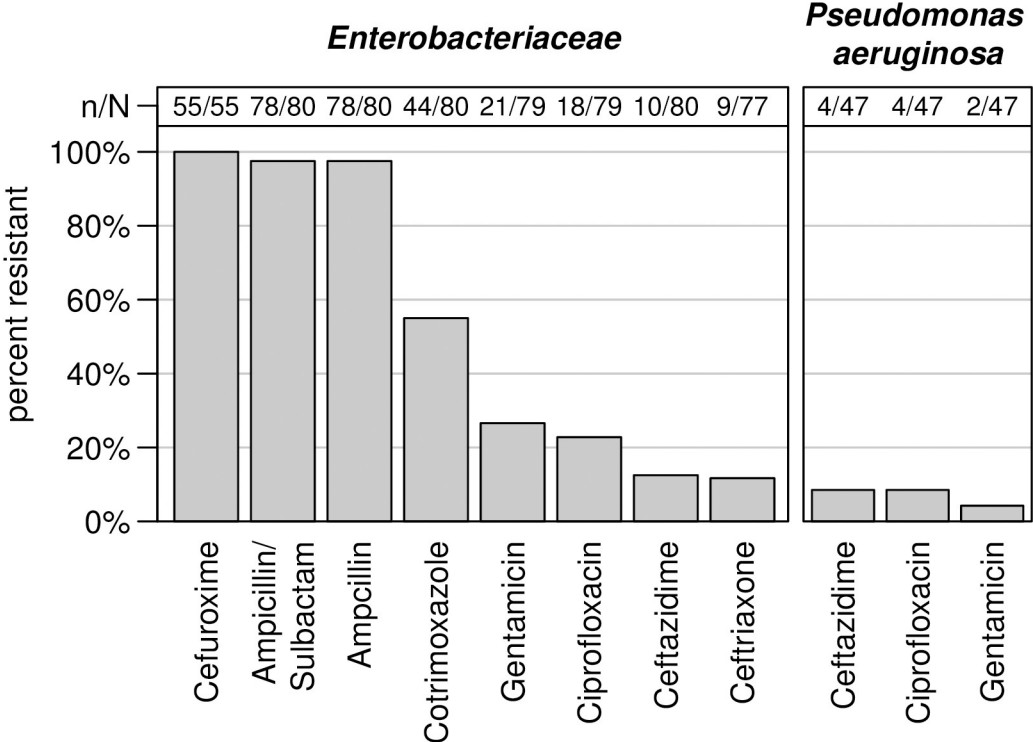

**Fig 2. Proportion of *Enterobacteriaceae* and *P. aeruginosa* resistant to locally available antibiotics.**

resistant *S. aureus* (MRSA). Inducible clindamycin resistance was found in 1/28 (4%) *S. aureus* isolate only. Extended-spectrum beta-lactamase (ESBL) producing *E. coli* and *Klebsiella pneumoniae* were 33% (n/N = 4/12) and 24% (n/N = 5/21), respectively. Fluoroquinolone resistance was seen in 58% (n/N = 7/12) *E. coli*, 14% (n/N = 3/21) *K. pneumoniae* and 9% (n/N = 4/47) *P. aeruginosa*. Two of *K. pneumoniae* isolates and one *E. coli* exhibited resistance to three antibiotic groups: penicillins, 3$^{rd}$ generation cephalosporin and fluoroquinolones. One *P. aeruginosa* strain exhibited resistance to the same groups of antibiotics in addition to resistance to carbapenems. Antibiotic resistance profiles of *Enterobacteriaceae* and *P. aeruginosa* are shown in Fig 2.

### Amplification of fungal DNA from chronic wound swabs

In 23 (22%) out of 105 sampled wounds, fungal DNA was detected by broad-range PCR. Most frequent fungi were *Candida tropicalis* (n = 6; 6%) and *Candida* spp. (n = 5; 5%). A complete list of isolated fungi is presented in Table 3. Patients with multiple bacterial infections had a higher risk to be infected or colonized with a mycotic isolate (PR = 2.4; 95%-CI: 1.0–5.3). The broad-range PCR did not amplify DNA of typical fungal pathogens causing chronic skin infections including chromoblastomycosis and mycetoma.

### Discussion

In this study, we assessed the microbial composition of chronically infected wounds in rural Ghana. The bacterial spectrum seen was in line with findings from previous studies conducted in another region of Ghana, in Ethiopia and in Tanzania [1, 11–15].

**Table 3. Fungal isolates detected in 105 patient samples tested by PCR.**

| Isolate | Frequency (%) |
|---|---|
| *Candida tropicalis* | 6 (6) |
| *Candida* spp. | 5 (5) |
| *Candida* mixed sequence | 4 (4) |
| *Candida albicans* | 3 (3) |
| *Cladophiolophora* spp. | 1 (1) |
| *Fusarium solani* | 1 (1) |
| *Geotrichium candidum* | 1 (1) |
| *Purpureocillium/Acremonium* | 1 (1) |
| *Zygosaccaromyces* spp. | 1 (1) |

Altogether, Gram-negative bacteria dominated quantitatively. *Enterobacteriaceae*, *P. aeruginosa* and *S. aureus* were the most frequently isolated organisms. Previous assessments both from America and Europe as well as from Africa confirm *S. aureus* and *P. aeruginosa* as the leading pathogens isolated from chronic wounds. Both species express virulence factors and surface proteins negatively affecting wound healing [1,9,16,17]. Pathogenicity in chronic wounds is further increased by co-infections of *S. aureus* and *P. aeruginosa*, which show synergistic interactions in in-vitro models [16,18].

Biofilms caused by *P. aeruginosa* play important roles by maintaining chronic wound infections thus preventing healing. Such biofilms are also known to show high adherence to biological surfaces, a phenomenon which limits the sensitivity of swabbing-based sampling with subsequent culture-based diagnostic approaches in comparison to molecular biological approaches [19]. Consequently, it is likely that the true prevalence of *P. aeruginosa* was even higher. In this study, histological assessment of biofilms on chronic wounds [19] was not performed. Accordingly, assumptions on the likely impact of biofilm formation remain speculative.

Although *P. aeruginosa* followed by *S. aureus* were the most common bacteria identified on species level, the most frequent isolates were *Enterobacteriaceae*. The estimation of their clinical relevance is much more difficult even though it is generally accepted that *Enterobacteriaceae* like *E. coli* and others might play a role in wound infections [17,20]. Inhabitants of tropical or subtropical climate zones were shown to have high colonization rates of skin and mucous membranes with Gram-negative rod-shaped bacteria [21]. High frequencies were also reported from patients, students, and health-care workers in Madagascar [22]. This makes the discrimination between wound colonization and wound infection challenging. From the less frequently isolated bacteria, beta-hemolytic streptococci including *Streptococcus pyogenes* are highly likely to be etiologically relevant for the assessed chronic wound infections [23], while enterococci are frequent colonizers of uncertain clinical relevance in wounds [24]. In mono- and polymicrobial infections, *P. aeruginosa* was the most common bacterial specie. *Enterobacteriaceae* and enterococci were rarely observed alone, so it is probable they are contamination flora or minor components of polymicrobial infection.

Overall antibiotic resistance was considerably high. This is particularly true for orally locally available drugs typically used for empiric treatment of infections. The recorded 29% MRSA is in discordance with a Ghanaian study on wound infections from 2014, where no MRSA were identified from infected wounds [14]. This suggests a considerable increase of MRSA within less than a decade; although it has to be considered that selection processes in chronic wounds after repeated antimicrobial therapy are likely. High frequencies of 2nd and 3rd generation cephalosporins but absence of carbapenem resistance in *Enterobacteriaceae* isolated from

wounds has been described before in previous studies from Ghana [14]. Further fluoroquinolone resistance is on the increase in Ghana [25], significantly reducing the value of this group of antibiotics for treatment of severe infections. In addition, we found one carbapenem-resistant *P. aeruginosa*. Carbapenem-resistance is still comparatively rare in Ghana: 2.9% in Gram-negative rod-shaped bacteria, predominantly in *P. aeruginosa* and *Acinetobacter baumannii* [26]. In summary, the observed resistance patterns make any rational antimicrobial therapy challenging, indicating a need for routine diagnostics including antibiotic susceptibility testing. However, antibiotic resistance seen in bacterial isolates from chronic wounds is likely to overestimate the true resistance rate in clinical isolates, because selection processes due to repeated attempts of antibiotic treatment have to be considered as likely.

In regards to fungi isolated from chronic wounds, information on the immune status of the patients was not assessed. Hence potential etiological relevance remains speculative. In general, fungal infections have been seen to cause skin infection including mycetoma that may manifest as chronic wounds after trauma [27]. Detection of causative fungal pathogens requires prolonged culture on fungus-specific media or demonstration of fungal elements in tissues. The performance of broad-range fungal PCR to amplify fungal DNA for the diagnosis of these infections has not been evaluated [28]. Our results suggest that the applied approach may be more likely to amplify DNA of yeasts that may be normal skin microbiota or microbial communities of chronic wounds. Specific PCR assays may be more sensitive for the amplification of specific fungal pathogens such as *Madurella mycetomatis* [27]. Limitations of this study include geographic restrictions of sampling, reducing the representativeness of the study for the whole of Ghana. In addition, swabbing of neighboring sites of intact skin to better differentiate infection from colonization [29] was not performed. Also, more sensitive molecular diagnostic methods were not applied, making it likely that several components of mixed infections will have been missed [19]. Also, despite their potential involvement in chronic wound infections, no anaerobic bacterial culture was performed. Lastly, sample transport time of up to six hours may have facilitated overgrowth of certain bacterial species, possibly masking the growth of less viable bacteria, even though cool boxes were used.

## Conclusions

As in other studies elsewhere, *P. aeruginosa* and *S. aureus* were the most frequently detected isolates and thus the most important species to be considered by physicians for therapeutic decisions.

In addition, high rates of antibiotic resistance are to be expected for isolates from chronic wounds in Ghana. This stresses the need for routine bacterial diagnostics including antimicrobial resistance testing prior to targeted antimicrobial therapy if antibiotic treatment is required and locally disinfecting or surgical procedures are considered insufficient. If routine bacteriology is impossible for logistical or infrastructural reasons, the observed lack of carbapenem-resistance suggests that antibiotic substances, which are available for intravenous application only, may still have good chances of showing clinical effect. However, decision-makers should be aware that empiric treatment further increases antibiotic resistance and is thus a less sustainable approach.

## Supporting information

**S1 Table. 207 Potential pathogenic bacteria isolated from 105 patient wounds.**
(DOCX)

## Acknowledgments

We are grateful to all patients, who participated in this study and to the personnel at the Agogo Presbyterian Hospital. Without their efforts, this research study would not have been possible.

## Author Contributions

**Conceptualization:** Ralf Krumkamp, Kwabena Oppong, Benedikt Hogan, Jürgen May, Denise Dekker.

**Data curation:** Ralf Krumkamp, Denise Dekker.

**Formal analysis:** Ralf Krumkamp, Ricardo Strauss, Anna Jaeger, Denise Dekker.

**Investigation:** Kwabena Oppong, Charity Wiafe-Akenten, Kennedy G. Boahen, Volker Rickerts, Ilka McCormick Smith, Uwe Groß, Marco Schulze, Nimako Sarpong, Denise Dekker.

**Methodology:** Benedikt Hogan, Ricardo Strauss, Charity Wiafe-Akenten, Volker Rickerts, Ilka McCormick Smith, Uwe Groß, Marco Schulze, Denise Dekker.

**Project administration:** Denise Dekker.

**Software:** Anna Jaeger.

**Supervision:** Kwabena Oppong, Benedikt Hogan, Kennedy G. Boahen, Nimako Sarpong, Ellis Owusu-Dabo, Denise Dekker.

**Validation:** Jürgen May.

**Visualization:** Denise Dekker.

**Writing – original draft:** Ralf Krumkamp, Hagen Frickmann, Volker Rickerts, Uwe Groß, Marco Schulze, Ulrike Loderstädt, Denise Dekker.

**Writing – review & editing:** Denise Dekker.

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
