## [Decision Letter · Decision Letter 0]

5 May 2020

PONE-D-20-08668

Spectrum of antibiotic resistant bacteria and fungi isolated from chronically infected wounds in a rural district hospital in Ghana

PLOS ONE

Dear Dr dekker,

Thank you for submitting your manuscript to PLOS ONE. After careful consideration, we feel that it has merit but does not fully meet PLOS ONE’s publication criteria as it currently stands. Therefore, we invite you to submit a revised version of the manuscript that addresses the points raised during the review process.

I have received the reviews of your manuscript. While your paper addresses an important question, the reviewers stated several concerns about your study and did not recommend publication in its present form.  All reviewers voice a number of concerns regarding the rational of why Ghana rural localization was chosen for such observation and how data analysis were done.  These comments need to be addressed carefully. In addition there were numerous issues identified where additional experimentation and documentation is needed.  Please see reviewers’ insightful comments below. Personally, at a more detailed level, I find the manuscript could benefit from strengthening the rationale of the study (see specific comments below).

Specific comments:

It is customary to abbreviate the name of the genus after the first time it is used, for examples, *Staphylococcus aureus *should be* S. aureus *after first mention in the paper (Ref:  International Code of Nomenclature of Bacteria: Bacteriological Code, 1990 Revision).It may be valuable to have brief description of the different stages of wounds to give readers a general idea since this was mentioned in the results.Line 59 – 61:  Move this sentence down as third paragraph and expand to explain what other fungal pathogens could cause problem.Line 72 – 73:  The rationale of the study needs to be strengthen further here*.*Line 87 – 90:  Need approval number.Line 229 – 234:  This begs the question on how to access whether there are biofilm formation on the open wound.Line 253 – 255:  This sentence is awkward, please rephrase for clarity.Line 335 – 340:  Reference 9&10 need space in between for format consistency.

We would appreciate receiving your revised manuscript by Jun 19 2020 11:59PM. To enhance the reproducibility of your results, we recommend that if applicable you deposit your laboratory protocols in protocols.io, where a protocol can be assigned its own identifier (DOI) such that it can be cited independently in the future. For instructions see: http://journals.plos.org/plosone/s/submission-guidelines#loc-laboratory-protocols

We look forward to receiving your revised manuscript.

Kind regards,

Baochuan Lin, Ph.D.

Academic Editor

PLOS ONE

Journal Requirements:

Reviewers' comments:

Reviewer's Responses to Questions

**Comments to the Author**

1. Is the manuscript technically sound, and do the data support the conclusions?

Reviewer #1: Yes

Reviewer #2: Partly

Reviewer #3: Yes

2. Has the statistical analysis been performed appropriately and rigorously? 

Reviewer #1: Yes

Reviewer #2: No

Reviewer #3: Yes

3. Have the authors made all data underlying the findings in their manuscript fully available?

Reviewer #1: Yes

Reviewer #2: Yes

Reviewer #3: Yes

4. Is the manuscript presented in an intelligible fashion and written in standard English?

Reviewer #1: Yes

Reviewer #2: Yes

Reviewer #3: Yes

5. Review Comments to the Author

Reviewer #1: Title: Spectrum of antibiotic resistant bacteria and fungi isolated from chronically infected

wounds in a rural district hospital in Ghana

The manuscript entitled “Spectrum of antibiotic resistant bacteria and fungi isolated from chronically infected wounds in a rural district hospital in Ghana “ described a pathogen distribution in chronic wounds in rural Ghana. It seems that information matched to the international patterns with a predominance of Pseudomonas aeruginosa and Staphylococcus aureus. Moreover, very high resistance rates were also observed suggestion the use or not of certain antimicrobials. Data here reported are important but major revisions are necessary before manuscript acceptance. Furthermore, some suggestions were here providing in order to improve the manuscript quality.

Suggestions:

1. Its is not clear why Ghana rural localization was chosen for such observation. For me it is obvious, specially because I could observe some Ghanaian authors. However, at introduction authors are invited to better explain spot location, adding such information in a World context.

2. In line 99, authors described that take 6 hours to transport the sample. I assume that collection was in a not easy place. But is this not a limitation? Authors could at least describe the possible contaminations occurring at this time.

3. In line 113, authors are invited to better describe the MALDI biotyper analyses including the number of spectra used in each identification and the number of technical and biological replicates utilized.

4. Please provide the list of antibiotics used for susceptibility tests. This could be added as supplementary material.

5. Please provide the primers sequence used at fungal identification.

6. There is some problem in Figure 1. The same occurs in figure 2, inserted in main text. Nevertheless at the end of main text is everything ok.

7. Table 2. Its is amazing that all strains were 100% resistant to CXM. This is not impossible but unusual. Authors are invited to clearly discussed. Moreover, is also remarkable that Klebsiella show so high rates of resistance. This is also an important data.

8. Why did authors did not test the fungal susceptibility to antifungals? Such data could also be important in order to determine the difficult in the region.

Reviewer #2: The objective of the study is to determine the epidemiology of microbial pathogens in chroncially infected wounds to support therapeutic choices for physician.

The aetiology of the chronic wounds in Table 1 is not welly defined (does cut means surgical incision or trauma?). Suggest for author to classify the types of wound by diabetic, arterial, infectious, surgical, trauma, venous and pressure ulcers.

Authors have performed logistic regression for wound size, traditional medicine and multiple bacterial infections. The p-value for the regression analysis are not stated in the manuscript.

Suggest to look into

1. Association of antibiotic usage and antibiotic resistance.

2. Association of types of wound and antibiotic resistance.

In row 211, authors have mentioned that "In 23(22%) wound, fungal DNA was amplified by broad range PCR". Does it means only 23 wounds were amplified for fungal detection instead of 105 wounds? Please justify.

Reviewer #3: Dear Editor,

Thank you for having chosen me as Referee for this paper.

I was pleased to read this work related to chronic infected wound in Ghana.

The authors provide detailed information on the epidemiology of microbial pathogens in chronically infected wounds in rural Ghana with the aim to support therapeutic choices for physicians. For the study were enrolled 105 patients with 207 microbial isolates. Isolated strains were characterized by a high resistant rate also against carbapenem. They concluded that for a correct management of these infections it is important to perform microbiological diagnostic approach including susceptibility testing.

The work is well structured and the topic, the antibiotic resistance of pathogen microrganisms isolated from chronically infected wounds in undeveloped countries such as Ghana, is of significant importance and fits within the scope of Plos one.

The results justify the conclusion.

All sections are presented with adequate clarity.

I suggest to emphasize the poor situation in Africa and in rural Ghana in the Introduction section.

In my opinion, the overall content of the paper is significant and the manuscript could be recommended for publication in “Plos one” after minor revision.

I have suggested some corrections and formulated some suggestions that might help the Authors in improving the manuscript.

Minor revision

In all parts of the manuscript, I recommend pointing the name of the bacteria except for the first time

• Ethics Statement and Ethical considerations (lines 86-90): Please, include the approval number of Ethical Committee

• Please, change “flora” with “microbiota” in all parts of the MS

• Page 7, lines 122-123, please insert “ATCC” before the number of the strains

• Page 7, line 150. Please, change 47 with 49

• Page 7 lines 160-163, Were the antimicrobial patterns of strains isolated from patients previously treated with antibiotics different from those of strains isolated from patients that used herbal treatments? Please insert a comment in Discussion

• Page 7, line 160. Please, add the space (n=11;10%)

• Page 12, line 228. Could be useful insert a comment regarding the following reference:

Synergistic interactions of Pseudomonas aeruginosa and Staphylococcus aureus in an in vitro wound model. DeLeon S, Clinton A, Fowler H, Everett J, Horswill AR, Rumbaugh KP. Infect Immun. 2014 Nov;82(11):4718-28. doi: 10.1128/IAI.02198-14. Epub 2014 Aug 25

• Page 12, lines 235-237 Please the sentence is not clear, please rephrase it

• Page 12, line 238. Please, add uppercase letter

• Page 12, line 239 delete full stop and insert comma

• Page 13, lines 255-260. Please, rephrase the sentences

• Page 13, line 259 delete full stop and insert comma

• Page 16, line 305, change number 3 with 1

• Figure 1 it is not clear, please, re-write the legend or modify the figure

6. PLOS authors have the option to publish the peer review history of their article (what does this mean?). If published, this will include your full peer review and any attached files.

Reviewer #1: No

Reviewer #2: No

Reviewer #3: Yes: Luigina Cellini, Dept of Pharmacy, "G. d'Annunzio" University, Chieti, Italy

Mara Di Giulio, Associate Professor, Dept of Pharmacy, "G. d'Annunzio" University, Chieti, Italy

Silvia Di Lodovico, PhD, "G. d'Annunzio" University, Chieti, Italy

---

## [Author Response · Author response to Decision Letter 0]

10 Jul 2020

Dear Editor and Reviewers,

Thank you for the valuable input and suggestions. Please find below our comments in italics. We have amended the manuscript accordingly and do hope that the changes are acceptable to you.

Kind regards,

Denise Dekker

---

## [Decision Letter · Decision Letter 1]

23 Jul 2020

Spectrum of antibiotic resistant bacteria and fungi isolated from chronically infected wounds in a rural district hospital in Ghana

PONE-D-20-08668R1

Dear Dr. dekker,

We’re pleased to inform you that your manuscript has been judged scientifically suitable for publication and will be formally accepted for publication once it meets all outstanding technical requirements.

Kind regards,

Baochuan Lin, Ph.D.

Academic Editor

PLOS ONE

Additional Editor Comments (optional):

Reviewers' comments:

Reviewer's Responses to Questions

**Comments to the Author**

1. If the authors have adequately addressed your comments raised in a previous round of review and you feel that this manuscript is now acceptable for publication, you may indicate that here to bypass the “Comments to the Author” section, enter your conflict of interest statement in the “Confidential to Editor” section, and submit your "Accept" recommendation.

Reviewer #1: All comments have been addressed

Reviewer #3: All comments have been addressed

2. Is the manuscript technically sound, and do the data support the conclusions?

Reviewer #1: Yes

Reviewer #3: Yes

3. Has the statistical analysis been performed appropriately and rigorously? 

Reviewer #1: Yes

Reviewer #3: Yes

4. Have the authors made all data underlying the findings in their manuscript fully available?

Reviewer #1: Yes

Reviewer #3: Yes

5. Is the manuscript presented in an intelligible fashion and written in standard English?

Reviewer #1: Yes

Reviewer #3: Yes

6. Review Comments to the Author

Reviewer #1: The manuscript was improved been acceptable in the present form. In that view I have no additional comments

Reviewer #3: This is a much improved version of the previously submitted manuscript.

The present revision is suitable for publication on Plos One

7. PLOS authors have the option to publish the peer review history of their article (what does this mean?). If published, this will include your full peer review and any attached files.

Reviewer #1: **Yes: **Octavio Luiz Franco

Reviewer #3: **Yes: **Luigina Cellini

---

## [Editor Report · Acceptance letter]

28 Jul 2020

PONE-D-20-08668R1 

Spectrum of antibiotic resistant bacteria and fungi isolated from chronically infected wounds in a rural district hospital in Ghana 

Dear Dr. Dekker:

I'm pleased to inform you that your manuscript has been deemed suitable for publication in PLOS ONE. Congratulations! Your manuscript is now with our production department. 

Kind regards, 

on behalf of

Dr. Baochuan Lin 

Academic Editor

PLOS ONE